# Different Proteomic Profiles Regarding Antihypertensive Therapy in Preeclampsia Pregnant

**DOI:** 10.3390/ijms25168738

**Published:** 2024-08-10

**Authors:** Caroline C. Pinto-Souza, Julyane N. S. Kaihara, Priscila R. Nunes, Moises H. Mastella, Bruno C. Rossini, Bruna Cavecci-Mendonça, Ricardo de Carvalho Cavalli, Lucilene D. dos Santos, Valeria C. Sandrim

**Affiliations:** 1Department of Biophysics and Pharmacology, Institute of Biosciences of Botucatu (IBB), São Paulo State University (UNESP), Botucatu 18618-689, SP, Brazil; caroline.cp.souza@unesp.br (C.C.P.-S.); j.kaihara@unesp.br (J.N.S.K.); priscila.nunes@unesp.br (P.R.N.); m.mastella@unesp.br (M.H.M.); 2Biotechnology Institute (IBTEC), São Paulo State University (UNESP), Botucatu 18618-687, SP, Brazil; bruno.rossini@unesp.br (B.C.R.); bruna.cavecci-mendonca@unesp.br (B.C.-M.); lucilene.delazari@unesp.br (L.D.d.S.); 3Center for the Study of Venoms and Venomous Animals (CEVAP), São Paulo State University (UNESP), Botucatu 18619-002, SP, Brazil; 4Department of Gynecology and Obstetrics, Faculty of Medicine of Ribeirao Preto, University of Sao Paulo (USP), Ribeirao Preto 14049-900, SP, Brazil; rcavalli@fmrp.usp.br

**Keywords:** preeclampsia, antihypertensive therapy responsiveness, proteomics, fibronectin, pregnancy-specific beta-1-glycoprotein 1, complement C4A, complement C4B

## Abstract

Preeclampsia (PE) is a hypertensive pregnancy syndrome associated with target organ damage and increased cardiovascular risks, necessitating antihypertensive therapy. However, approximately 40% of patients are nonresponsive to treatment, which results in worse clinical outcomes. This study aimed to compare circulating proteomic profiles and identify differentially expressed proteins among 10 responsive (R-PE), 10 nonresponsive (NR-PE) patients, and 10 healthy pregnant controls (HP). We also explored correlations between these proteins and clinical data. Plasma protein relative quantification was performed using mass spectrometry, followed by bioinformatics analyses with the UniProt database, PatternLab for Proteomics 4.0, and MetaboAnalyst software (version 6.0). Considering a fold change of 1.5, four proteins were differentially expressed between NR-PE and R-PE: one upregulated (fibronectin) and three downregulated (pregnancy-specific beta-1-glycoprotein 1, complement C4B, and complement C4A). Between NR-PE and HP, six proteins were differentially expressed: two upregulated (clusterin and plasmin heavy chain A) and four downregulated (apolipoprotein L1, heparin cofactor II, complement C4B, and haptoglobin-related protein). Three proteins were differentially expressed between R-PE and HP: one downregulated (transthyretin) and two upregulated (apolipoprotein C1 and hemoglobin subunit beta). These findings suggest a complex interplay of these proteins involved in inflammatory, immune, and metabolic processes with antihypertensive therapy responsiveness and PE pathophysiology.

## 1. Introduction

Preeclampsia (PE) is a pregnancy-related hypertensive disorder that develops after 20 weeks of gestation. It is characterized by damage to multiple organs, primarily the liver, kidneys, and brain, as well as the presentation of other symptoms, including proteinuria, coagulation disturbances, pulmonary edema, and nervous system manifestations [1]. This clinical condition is associated with abnormal placentation, which results in fetal growth restriction, low birth weight, and premature birth in the newborn. Consequently, this disorder is associated with an elevated perinatal and maternal mortality and morbidity rate on a global scale, with an estimated prevalence of 2-8% in pregnancies worldwide [1].

Despite the global health implications of PE, the available treatment options remain limited. Current recommendations include the use of aspirin and calcium as preventive measures [1]. Moreover, antihypertensive therapy is indicated for the management of hypertension during pregnancy to delay delivery and reduce the incidence of complications associated with PE [2]. This pharmacological approach primarily involves the prescription of methyldopa, nifedipine, hydralazine, and labetalol [2]. However, it is notable that approximately 40% of patients with PE do not respond to antihypertensive therapy. This lack of response often leads to the least favorable clinical outcomes related to the pathophysiology of PE [3].

Our group has previously demonstrated that plasma from nonresponsive and responsive patients may differentially modulate the expression of genes related to endothelial cell biology. This evaluation focused on the effects of plasma from these patients on endothelial gene expression, thereby highlighting the distinct impact on endothelial cell function based on the patient’s responsiveness to treatment [3]. Furthermore, our group has shown that single-nucleotide polymorphisms (SNPs) are associated with antihypertensive therapy responsiveness in PE. In nonresponsive PE patients, the ARG2 SNP rs3742879 was associated with a reduction in arginase 2 levels and an increase in the formation of nitric oxide (NO), the pathway of which is often impaired in PE [4]. Likewise, it has been shown that nicotinamide phosphoribosyltransferase (NAMPT) SNP rs1319501 influences NO concentrations in nonresponsive PE patients. These findings suggest that NAMPT polymorphisms may affect plasma visfatin/NAMPT levels in nonresponsive PE patients [5].

The precise mechanisms responsible for the responsiveness of patients with PE to antihypertensive therapy remain unclear. Hence, proteomics represents a promising avenue that may offer valuable insights into the underlying causes of this failure in regulating blood pressure. Proteomics analysis relies on two principal approaches: targeted and untargeted approaches. The targeted approach focuses on identifying and quantifying specific known proteins, whereas the untargeted approach offers a comprehensive global analysis of the entire proteome, enabling the discovery of novel biomarkers [6]. Previous studies have compared the proteome of patients with PE, identifying multiple proteins that may be associated with its pathophysiology and serve as potential prediction biomarkers. These include differentially expressed membrane proteins [7], transport proteins [8], proteins involved in lipid metabolism, coagulation factors, the complement system [9], and pregnancy-specific glycoproteins [10]. Notably, these proteins are particularly significant in early-onset PE, a subtype associated with greater placental injury and preterm delivery. Despite this, to our knowledge, no studies have yet evaluated the proteome of this hypertensive disorder of pregnancy concerning its responsiveness to antihypertensive drugs. Therefore, this study aimed to perform an untargeted shotgun proteomic analysis of plasma proteins among pregnant women with PE who respond to antihypertensive therapy (R-PE) and those who do not (NR-PE) and with the healthy pregnant (HP) control. Additionally, this study aimed to identify potential correlations between the differentially expressed proteins and clinical data to explore how these proteins are associated with the pathophysiology of PE.

## 2. Results

Table 1 presents a summary of the characteristics of PE patients classified as R or NR to antihypertensive therapy and HP. The HP group had lower body mass index (BMI) at blood sampling and systolic blood pressure (SBP) at blood sampling, with these observations being consistent across both PE subtypes. Additionally, R-PE exhibited lower diastolic blood pressure (DBP) than the R-PE group. The lower blood pressure mean values observed in the R-PE group resulted from effective antihypertensive therapy after the initial diagnosis of PE, demonstrating the R-PE responsiveness to this treatment. The NR-PE group exhibited higher plasma urea levels and proteinuria levels than the R-PE group, as well as lower gestational age (GA) at delivery and newborn weight than the other groups (all *p* < 0.05). At the time of blood sampling, 60% of subjects in the R-PE group and 80% of subjects in the NR-PE group were taking methyldopa. Additionally, 30% of subjects in the NR-PE group were also taking nifedipine, while 10% were taking hydralazine.

Figure 1, Figure 2 and Figure 3 demonstrate the statistical methods applied to explore data related to the differentially expressed proteins in the following comparisons, respectively, R-PE versus NR-PE (Figure 1), HP versus NR-PE (Figure 2), and HP versus R-PE (Figure 3). Although Figure 1a and Figure 3a indicate minimal overlap between groups in the two-dimensional partial least squares discriminant analysis (PLS-DA) score plots, there are notable separations between all groups in Figure 2a and in Figure 1b, Figure 2b and Figure 3b via the three-dimensional sparse partial least squares discriminant analysis (sPLS-DA). This suggests that the differentially expressed proteins may drive the group separation and classification.

Regarding R-PE versus NR-PE, chemometric statistical analyses of the proteomics data using PLS-DA showed that the first component accounted for 15.0% of the total variation between the R-PE and NR-PE groups, while the second component was responsible for 11.2% (Figure 1a). Moreover, the sPLS-DA revealed that component 1 was responsible for 7.2% of the total between-group variation, component 2 accounted for 9.9%, and component 3 accounted for 8.3% (Figure 1b). Figure 1c displays pregnancy-specific beta-1-glycoprotein 1 (PSG1), fibronectin (FN1), and haptoglobin (HP) among the top three proteins that mostly contributed to the differentiation between the R-PE and NR-PE groups identified through PLS-DA, with VIP scores > 2.5. In Figure 1d, the volcano plot unveiled four differentially expressed proteins in the subgroups NR-PE and R-PE, three downregulated (PSG1, complement C4B, and complement C4A) and one upregulated (FN1). Figure 1e–h display the normalized data of the peak intensities of PSG1 (<NR-PE), C4B (<NR-PE), C4A (<NR-PE), and FN1 (>NR-PE), respectively.

Figure 2a displays the PLS-DA, indicating that 14.4% of the total variation between the HP and NR-PE groups was accounted for by the first component, while the second component was responsible for 8.7%. Moreover, the sPLS-DA revealed that component 1 was responsible for 8.5% of the total between-group variation, component 2 accounted for 9.6%, and component 3 accounted for 10.2% (Figure 2b). Figure 2c displays HP, apolipoprotein L1 (APOL1), and heparin cofactor II (SERPIND1) among the top three proteins that mostly contributed to the differentiation between the HP and NR-PE groups identified through the use of PLS-DA, with VIP scores > 2.5. In Figure 2d, the volcano plot unveiled six differentially expressed proteins in NR-PE regarding HP: four downregulated (APOL1; SERPIND1; C4B; and HPR, haptoglobin-related protein) and two upregulated (CLU, clusterin; and PLG, plasmin heavy chain A). Figure 2e–j display the normalized data of the peak intensities of CLU (>NR-PE), APOL1 (<NR-PE), SERPIND1 (<NR- PE), PLG (>NR-PE), C4B (<NR-PE), and HPR (<NR-PE), in this order.

The PLS-DA between the HP and R-PE groups showed that the first component was responsible for 10.0% of the total between-group variation, while the second component was responsible for 15.6% (Figure 3a). Additionally, in Figure 3b, the sPLS-DA revealed that component 1 accounted for 8.5% of the total variation between the groups, component 2 accounted for 9.9%, and component 3 accounted for 8.3%. Figure 3c displays apolipoprotein C1 (APOC1) and PSG1 among the top two proteins that mostly contributed to the differentiation between the R-PE and HP groups identified through PLS-DA, with VIP scores of >2.5. Figure 3d shows a comparison between R-PE and HP using a volcano plot, indicating three differentially expressed proteins considering a fold change of 1.5: one upregulated (TTR, transthyretin) and two downregulated (APOC1 and HBB, hemoglobin subunit beta). Figure 3e–g display the normalized data of the peak intensities of APOC1, HBB, and TTR, respectively. 

Next, we correlated the different expressions and the clinical characteristics of PE patients, encompassing both R-PE and NR-PE (Table 2). The parameters of SBP and DBP and creatinine exhibited a negative correlation with the differential expression of C4A and C4B (respectively r = −0.5199; r = −0.5193, r = −0.4919, r = −0.4916, rs = −0.5969, r = −0.5833). Conversely, the expression of these two proteins showed a positive correlation with GA at delivery and newborn weight, respectively, rs = 0.6782 and rs = 0.6775, rs = 0.6315, and rs = 0.6316. The differential expression of FN1 was found to be positively correlated with serum glutamic oxaloacetic transaminase (SGOT, r = 0.4648), while PSG1 demonstrated a positive correlation with GA at delivery (rs = 0.6013) and newborn weight (rs = 0.6508).

## 3. Discussion

To our knowledge, this is the first study to explore proteomic profiles in plasma samples from patients with PE in the context of responsiveness to antihypertensive therapy. This study represents a pioneering contribution by demonstrating the differential protein profile in the plasma of R-PE and NR-PE women, an observation not previously reported in the scientific literature. The absence of previous investigations makes it difficult to directly correlate with previous studies. Given the lack of comparative data, our study is positioned as an initial milestone that can serve as a basis for subsequent investigations, encouraging a reassessment of the functions and behaviors of proteins in the pathophysiology of PE. Considering the NR-PE and R-PE subgroups, we observed four differentially expressed proteins. This study sheds light on important proteins such as FN1, those of the complement system (C4B and C4A), and PSG1 when comparing the two PE subgroups.

Our data indicate that circulating levels of FN1 are upregulated in the NR-PE group (Figure 1), which agrees substantially with findings from prior studies that have demonstrated increased levels of this protein in preterm delivery (which exhibits symptoms and risk factors that are similar to PE) compared to term delivery [11,12] in women who develop the disorder versus healthy pregnancy [13,14], as well as in gestational diabetes and obesity [12]. In the pathophysiology of PE, FN1 is involved in several molecular mechanisms that contribute to the development and progression of the disease, such as vascular remodeling and angiogenesis, inflammation and modulation of the immune response, oxidative stress, endothelial dysfunction, dysregulation of growth factors, and the restructuring of the extracellular matrix; furthermore, the clinical pathophysiology of PE may be related to dysfunctional FN1 plasma levels [15].

The role of FN1, a glycoprotein synthesized by the *FN1* gene, has been well described in many studies [16,17,18], and its principal role can be summarized as a mediator of tissue remodeling. Due to its location within the extracellular matrix (ECM), FN1 is involved in the remodeling of the tissue microenvironment, which is essential for processes such as adhesion, cell migration, and wound healing. Consequently, FN1 exerts a key role in the inflammatory process [19]. It is important to emphasize that FN may be present in both plasma and cellular form, and despite both contributing to the ECM, they have different chemical and biological properties [20]. Hypoxic environments can affect the ECM and also increase FN levels. This stimulus is crucial for placental development and fetal growth; however, chronic hypoxia associated with high plasma FN levels has been implicated in the pathogenesis of PE [21]. Some authors have proposed that a high level of FN in plasma may serve as a biomarker of PE and as a means of differentiating this disorder from gestational hypertension. This hypothesis is supported by the correlation between FN and endothelial injury, both of which resolve after delivery [22].

The FN1’s ability to shape the tissue may be either directly or indirectly modulated. One example is related to the process of cell death. Previous reports found that elevated levels of FN1 induce apoptosis and autophagy in human umbilical vein endothelial cells (HUVECs) supplemented with plasma from PE patients by inhibiting the cascade related to the mammalian target of rapamycin (mTOR). The subsequent silencing of FN1 has been shown to restore the pathway. Conversely, oxidative stress scenarios are linked to pregnancy complications, such as PE [23], and can directly impact the FN1 form. The imbalance resulting from the excessive production of reactive oxygen species (ROS) may occur due to a reduction in the efficiency of antioxidant enzymes, such as superoxide dismutase 2 (SOD2), which are responsible for converting superoxide into hydrogen peroxide (H_2_O_2_). This enzyme was associated with FN dimerization, polymerization, and location outside of the cell. During the migration process, when mitochondria move to the edge of the cells and the FN is important to the orientation, lower signaling by H_2_O_2_ may affect the stabilization of FN [24], contributing to worsening outcomes in gestational disorders. This is because of mitochondrial dysfunction and, consequently, oxidative stress [25], which contribute to hypertension conditions.

In contrast, the C4B and C4A proteins from the complement system were downregulated in the NR-PE relative to the R-PE (Figure 1), a finding that aligns with previous proteomic studies [10,26,27,28,29]. Nonetheless, to our knowledge, this is the first study to elucidate the expression of these proteins in these two PE subgroups, thereby paving the way for further investigation of these proteins and the activation of the complement system in pregnant women with PE. Furthermore, the peak intensities of C4B and C4A presented a concordant biological response with clinical variables, such as positive correlations with newborn weight and GA at delivery, as well as negative associations with both blood pressure and creatinine levels in PE (Table 2). Studies have also demonstrated that women with PE have significantly lower circulating levels of C4 compared to normotensive pregnancies. The reduction in circulating levels of C4 may be a signal of consumption of complement factors due to systemic activation of the immune response [30,31]. One potential mechanism by which C4 may act is through activation fragments, which can modulate immune reactions to restrain hyperinflammatory reactions induced by cytokines [32]. C4 proteins participate in the classical and mannose-binding lectin complement activation pathways [33].

The regulation of the complement system is mediated by several circulating and cell membrane proteins. Many of these proteins are members of a family of proteins known as regulators of complement activity (RCA). C4b deposited on cell surfaces is bound by DAF (Decay-Accelerating Factor), CR1 (Complement Receptor Type 1), MCP (Complement Receptor Type 1), and another plasma protein, C4BP (C4 Binding Protein) [34]. Regulation entails the prevention of complement activation and the degradation of the proteins implicated in this cascade [35]. In a study conducted by Lokki and colleagues, placental C4BP was identified in 80% of the syncytiotrophoblast from control subjects, whereas 42% of the preeclamptic placentas exhibited some degree of C4BP deposition in the syncytiotrophoblast layer [36].

It is becoming increasingly clear that inflammation and immunity are essential in the development of hypertension. Indeed, there is growing evidence that complement activation-mediated innate immune response plays a part in regulating hypertension and target organ damage. Nevertheless, it remains uncertain whether complement can influence the elevation of blood pressure and the development of hypertension [37]. The downregulation of C4 in NR-PE and the negative correlation with BP can be explained as a compensatory mechanism present in the pathophysiology of PE, which is still far from being fully elucidated. The subgroup of NR-PE may experience adverse clinical outcomes due to their nonresponsiveness to antihypertensive therapy [38] and factors such as antihypertensive therapy, the presence of autoantibodies (e.g., angiotensin II type I auto-antibody, already described in PE) [39], and the relationship between complement activation and the regulation of the immune system of these patients can be decisive in the search for new study targets and biomarkers for PE.

Similarly, PSG1 is also downregulated in the NR-PE subgroup (Figure 1), and its levels exhibited a positive correlation with newborn weight and GA at delivery (Table 2). This protein is closely associated with the development of PE [40]. Likewise, PSG-1 is well documented in molecular pathways that are involved in blood pressure regulation [41]. Toprak et al. suggested that low PSG-1 contributes to the disruption of circadian blood pressure regulation [42]. Among the 11 different *PSG* genes, PSG1 is the most abundantly secreted molecule during early pregnancy and from term placenta [43]. It is secreted by trophoblast cells and increases in concentration throughout pregnancy, becoming one of the most abundant proteins in maternal serum by the third trimester [44]. In this case, the low levels of PSG1 found in nonresponsive patients may be explained by the fact that these patients generally have worse outcomes, such as preterm birth, low placental weight, and low newborn birth weights. Together with other protein family members, it may also have protective functions in an oxidative stress context. Although PSG1 does not appear to affect growth or morphology in non-endothelial cells, it has been demonstrated to confer resistance to certain harmful agents that lead to DNA single-strand breaks, thereby reducing this kind of damage during pregnancy [45]. In light of these findings, PSG levels have been linked with maintaining a healthy gestation [40,46,47]. However, it is important to note that the amount of PSG1 can vary considerably depending on the methodology employed in a given study, with some studies reporting lower levels of PSG1 [40] and others reporting higher levels [29]. For example, GA has been identified as a possible interfering factor in the protein concentration and, thus, considered a limitation by some authors. PSG1 is associated with tube formation in endothelial cells; however, this event can be induced by other PSGs [48]. This role leads us to suggest that the lower expression of the protein observed in the NR-PE group may be associated with signaling pathways that activate compensatory mechanisms in an attempt to enhance blood flow [49,50] and the provision of an immunomodulatory environment at the maternal–fetal interface indicates that this protein family is equally effective in successful pregnancy outcomes [51]. 

The comparison between NR-PE and HP (Figure 2) reveals the upregulated expression of CLU and PLG, accompanied by the downregulated expression of APOL1, SERPIND1, C4B, and HPR. A study conducted by Zeng and colleagues indicated that CLU may be involved in the regulation of trophoblast invasion and migration during placental development, which may be one of the risk factors for PE [52]. Likewise, other studies have demonstrated that elevated levels of CLU are linked to intrauterine growth restriction [53] and are markedly upregulated in women with severe PE [54]. PLG1 has been extensively investigated in the context of renal complications as it plays roles in disorders afflicted with proteinuria and sodium imbalance, such as diabetes, PE, and nephrosis [55]. SERPIND1 participates in the coagulation process [56], and the involvement of this protein in the inflammatory response helps to understand its inverse correlation with proteinuria levels. Following a similar logic, the immune implication of HPR [57] agrees with its inverse correlation with creatinine levels. Nonetheless, further studies are required to ascertain the participation of these differentially expressed proteins in the pathophysiology of PE. 

Regarding APOC1 and APOL1, both proteins are mainly produced in the liver and exhibit a reduction in the PE group (both R and NR, Figure 2 and Figure 3). The existing literature on the relationship between these proteins and preeclampsia is limited, underscoring the importance of further research to elucidate the role of APOC1 and APOL1 in the pathophysiology of preeclampsia and their potential use as biomarkers or therapeutic targets. Additionally, TTR was found to be upregulated in the R-PE. TTR is an acute-phase negative protein, a marker of inflammation, and presents an important role in the female reproductive system [58]. Our data diverge from previous studies that reported higher levels of TTR in PE versus control [8,59]. These discrepancies may be attributed to the methodologies employed by the investigators, including the use of specific protocols such as immunoassays, densitometric and nephelometric analyses in vivo, in vitro, and mouse models, among others. Additionally, the study populations exhibit specific clinical characteristics. Consequently, the time point of blood sampling may be a contributing factor, as the protein is more highly expressed during the initial trimester of pregnancy. Furthermore, TTR regulation differs according to the tissue source, which highlights the potential value of investigating its concentration in the uterus and placenta throughout gestation [58].

One limitation of our study is that blood samples for protein analysis were collected after the majority of participants in the study initiated pharmacological treatment. This raises the concern that the observed differentially expressed proteins may be influenced by the medication rather than solely reflecting how individuals respond to the treatment itself. It is possible that antihypertensive drugs may impact various biological processes, which could potentially affect the proteomic profiles that we have observed. However, it is worth noting that there is currently no literature directly linking these medications to the proteins we identified in our study. This highlights the necessity for further investigation to elucidate the understanding of how antihypertensive therapies impact and interplay with patient proteomic profiles, particularly in the context of PE.

## 4. Materials and Methods

### 4.1. Selection of Participants

A total of 20 PE patients, 10 R-PE and 10 NR-PE, were selected for this study, in addition to the 10 HP control group. This project was approved by the Research Ethics Committee of the Ribeirao Preto Medical School, University of Sao Paulo (FMRP-USP) for all human investigations (CAAE-37738620.0.0000.5440, 19 October 2020), and the participants were consecutively enrolled in the Department of Gynecology and Obstetrics of the University Hospital of FMRP-USP. Furthermore, the recruited participants provided their informed consent to participate in the study. The diagnosis of PE was made following the American College of Obstetricians and Gynecologists guidelines [1]. The participants were identified as having PE: hypertension after 20 weeks of gestation (systolic blood pressure ≥ 140 mmHg and diastolic pressure ≥ 90 mmHg in two blood pressure measurements at rest, 4 h apart), and proteinuria (≥0.3 g/L in 24-h urine or a dipstick reading of 2+) [1]. In the absence of proteinuria, PE is diagnosed as hypertension associated with thrombocytopenia (platelet count < 100 × 10^9^/L), impaired liver function (elevated blood concentrations of liver transaminases to twice the normal concentration), renal failure, pulmonary edema, and cerebral or visual symptoms [1]. Women with preexisting hypertension, with or without superimposed PE, or HELLP (hemolysis, elevated liver enzymes, and low platelet count) syndrome were excluded from this study. 

### 4.2. Antihypertensive Treatment and Drug Response Evaluation

Following the PE diagnoses and antihypertensive treatment, the clinical and laboratory parameters were evaluated to classify and categorize PE patients based on their antihypertensive responsiveness, as previously reported [3,4,5,60]. Methyldopa (1000–1500 mg/day), a centrally acting alpha-2 adrenergic agonist, was used as the antihypertensive of choice since it has been demonstrated to decrease blood pressure [2]. Furthermore, according to the National High Blood Pressure Education Program Working Group (NHBPEP), methyldopa is not teratogenic and is safe for use during pregnancy, based on prior research [61]. In cases where patients exhibited no significant response to methyldopa, nifedipine (40–60 mg/day) was administered. Hydralazine (5–30 mg) was only administered in cases of hypertensive crisis. Patients were monitored for any indications of PE, with fetal surveillance and laboratory tests conducted at least once a week. 

The classification of pregnant women with PE as NR to antihypertensive therapy was determined upon the conclusion of antihypertensive treatment and delivery based on the presence of one or more of the established criteria previously published [3,4,5,60]:Clinical symptoms include blurred vision, persistent headache or scotomata, and persistent right upper quadrant or epigastric pain;Systolic blood pressure > 140 mmHg and diastolic blood pressure > 90 mmHg, assessed using the blood pressure curve;Hemolysis, elevated liver enzymes, and low platelet count syndrome or proteinuria > 2.0 g/24 h;Creatinine > 1.2 mg/100 mL or blood urea nitrogen > 30 mg/100 mL; aspartate aminotransferase > 70 U/L and alanine aminotransferase > 60 U/L;Fetal hypoactivity or nonreactive fetus revealed by cardiotocography; intrauterine growth restriction, oligohydramnios, abnormal biophysical profile score, and Doppler velocimetry abnormalities, evaluated using ultrasound.

### 4.3. Plasma Sample Collection

A total of 15 mL of venous blood was collected from the study participants under the established standard procedures. The samples were drawn into Vacutainer tubes (Becton, Dickinson and Company, Franklin Lakes, NJ, USA) containing EDTA. These tubes were then centrifuged at room temperature for 10 min at 3200× *g*. Aliquots of 250 μL were separated and stored at −80 °C until use. Plasma samples were utilized for biochemical and proteomic analyses. For proteomic analyses, the sample selection included patients with similar clinical characteristics: white skin color, age, non-smokers, and non-primiparous women. This approach helped to minimize the potential influence of confounding factors associated with the development of the disorder.

### 4.4. Proteomics Analysis

#### 4.4.1. Sample Preparation

The samples were prepared by mixing 2 μL of each crude sample with 98 μL of ammonium bicarbonate (Ambic, Sigma-Aldrich, San Luis, CA, USA). This proportion was scaled up by a factor of 7.5, and 15 μL of each sample was combined with 735 μL of Ambic to yield a final volume of 750 μL (1:50) suitable for protein quantification, electrophoresis, and digestion. The concentrated blood plasma was diluted using a sample homogenizer and centrifugation. After preparation, the samples were stored in the freezer until use.

#### 4.4.2. Protein Quantity and Quality

The proteins present in the plasma samples of PE subtypes were quantified via the Bradford method [62] using the Bio-Rad^®^ Protein Assay kit (code 500-0001, Bio-Rad Laboratories Inc., Hercules, CA, USA), with bovine albumin (BSA, Sigma-Aldrich, San Luis, CA, USA) serving as the standard protein. Subsequently, the quality and viability of the protein samples were assessed through one-dimensional electrophoretic analyses under reducing and denaturing conditions (SDS-PAGE), as previously described [63]. 

#### 4.4.3. Enzymatic Digestion of Proteins in Solution

Firstly, 50 μg of total proteins from each pool were diluted in 60 μL of 50 mM Ambic. Each pool was then incubated at 37 °C for 60 min with the addition of 25 μL of RapiGest SF Standard surfactant (code 186001861, Waters Corporation, Milford, CT, USA). Next, the pools were subjected to reduction and alkylation steps, utilizing 10 mM dithiothreitol (Sigma-Aldrich, San Luis, CA, USA) and 45 mM iodoacetamide (Sigma-Aldrich, San Luis, CA, USA), respectively. Enzymatic digestion in solution was conducted by applying the enzyme trypsin at a concentration of 1:100 (enzyme/sample) solubilized in 50 mM Ambic buffer at pH 7.8. Hydrolysis proceeded for 18 h and was interrupted with the addition of 10 μL of 5% (*v*/*v*) formic acid.

Then, the pools were incubated for 90 min at room temperature and centrifuged at 14,000× *g* at 6 °C for 30 min. The supernatant was removed and transferred to a new tube. The tryptic digests from each sample were subjected to peptide desalination columns (Cleanup C18 Spin, code 5188-2750, Agilent Technologies, Santa Clara, CA, USA) following the manufacturer’s instructions.

#### 4.4.4. Peptide Sequencing by Mass Spectrometry (MS)

MS analyses were executed using an Ultimate 3000 LC liquid nanochromatography equipment (Dionex, Germering, Germany) coupled to Q-Exactive mass spectrometry equipment (Thermo Fisher Scientific, Bremen, Germany). The mobile phases used were as follows: (A) 0.1% [*v*/*v*] formic acid in LCMS water and (B) 0.1% [*v*/*v*] formic acid in 80% [*v*/*v*] acetonitrile. The peptides were loaded into a C18 pre-column, 30 μm × 5 mm (code 164649, Thermo Fisher Scientific, Bremen, Germany), and desalted in an isocratic gradient of 4%B for three minutes at a 300 nL/min flow rate. Subsequently, the peptides were fractionated via an analytical column (Reprosil-Pur C18-AQ, 3 μm, 120 Å, 105 mm, code 1PCH7515-105H354-NV, PicoChip, Bath, England) using a linear gradient of 4–55%B for 30 min, 55% at 90%B for one minute, maintained at 90%B for five minutes and re-equilibrated at 4%B for 20 min at a 300 nL/min flow rate. Ionization was obtained via a Nanospray ion source (PicoChip, Bath, England), and the mode of operation was positive ionization using the DDA method. 

MS spectra were acquired across a mass range of *m*/*z* 200 to *m*/*z* 2000, a resolution of 70,000, and an injection time of 100 ms. The fragmentation chamber was conditioned with collision energy between 29 and 35% with a 17,500 resolution, a 50 ms injection time, a 4.0 *m*/*z* isolation window, and 10s dynamic exclusion. The spectrometric data were acquired using Thermo Xcalibur software (RRID: SCR_014593, version 4.0.27.19, Thermo Fisher Scientific, Bremen, Germany).

#### 4.4.5. Analysis of Proteomics Data

The raw EM data in the RAW format were submitted to PatternLab software (version 4.0.0.84) [64] for protein identification. The main parameters employed in this tool were as follows: the UniProt database (RRID: SCR_002380, Taxonomy Homo sapiens), trypsin enzyme, the allowance of two missed cleavages, the post-translational modification carbamidomethylation of cysteine residues, the variable post-translational modification oxidation of methionine residues, and an MS and MS/MS tolerance error of 0.0200 ppm. The maximum FDR (False Discovery Rate) was considered to be ≤1%. A matrix compatible with the MetaboAnalyst program (RRID: SCR_015539, https://www.metaboanalyst.ca/, accessed on 22 July 2024) was constructed from the proteomic data using the spectral counts of each characterized protein, which were normalized for each protein by the weighted average of the technical triplicates of each sample [65]. Proteins identified in less than 70% of the pools were excluded from the subsequent analysis. Chemometric and univariate analyses were conducted using online software version 6.0 of MetaboAnalyst. Proteins common to all patients were analyzed using PLS-DA, sPLS-DA, VIP score plot, and heatmaps to identify the proteins responsible for the observed differences between the groups (scores > 1.0). Additionally, a volcano plot was constructed to indicate large-magnitude fold changes (y-axis, FC < −1.5 or >1.5) and statistical significance (x-axis, raw *p*-value = 0.05). The original mass spectrometry data presented in the study are openly available in the MassIVE Repository from Computer Science and Engineering University of California, San Diego (https://massive.ucsd.edu/, accessed on 22 July 2024) with the dataset identifier: MSV000095404, and https://doi.org/10.25345/C50V89V22, accessed on 22 July 2024.

### 4.5. Statistical Analysis

The clinical characteristics of the HP group and women with PE who were R or NR to antihypertensive therapy were compared using an unpaired *t* test, One-Way ANOVA, followed by Tukey’s multiple comparison tests for parametric data, or Kruskal–Wallis test, followed by Dunn’s multiple comparison tests for nonparametric data. Categorical variables were compared employing χ^2^ tests, while differentially expressed proteins and clinical data were compared using Pearson or Spearman correlation tests, as appropriate. The statistical analyses were performed using GraphPad Prism 9.5 (RRID: SCR_002798, GraphPad Software, San Diego, CA, USA). A *p*-value < 0.05 was considered statistically significant.

## 5. Conclusions

In conclusion, our study provides valuable insights into the differential proteomic profiles of PE patients, specifically R-PE and NR-PE, compared to HP. By employing MS and comprehensive bioinformatics analyses, we were able to identify several proteins, including fibronectin, pregnancy-specific beta-1-glycoprotein 1, complement C4B, complement C4A, clusterin, plasmin heavy chain A, apolipoprotein L1, heparin cofactor II, haptoglobin-related protein, transthyretin, apolipoprotein C1, and hemoglobin subunit beta as potential biomarkers of antihypertensive therapy responsiveness. These findings suggest the complex interplay of these proteins involved in inflammatory, immune, and metabolic processes with antihypertensive therapy responsiveness and PE pathophysiology, reinforcing the requisite for further research to elucidate these mechanisms and develop personalized therapies.

## Figures and Tables

**Figure 1 ijms-25-08738-f001:**
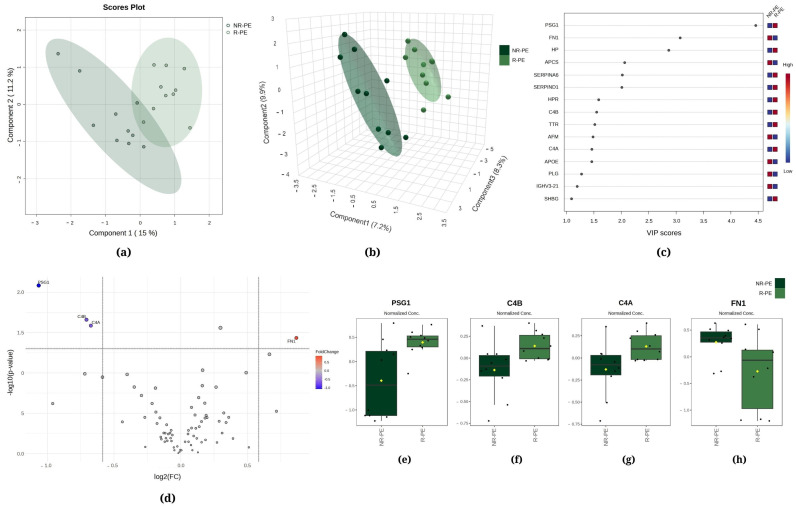
Chemometric and univariate statistical analyses of the proteins identified in the plasma of nonresponsive (NR-PE, in dark green) and responsive (R-PE, in light green) preeclampsia patients. (**a**) Two-dimensional partial least squares discriminant analysis (PLS-DA) score plot. (**b**) Three-dimensional sparse partial least squares-discriminant analysis (sPLS-DA) score plot. (**c**) Variable importance in projection (VIP) represents the score of the proteins that mostly contribute to the differentiation between the NR-PE and R-PE groups, as identified through sPLS-DA. (**d**) The volcano plot of the proteins showed four differentially expressed proteins when comparing the NR-PE with the R-PE group: three downregulated (PSG1, C4B, and C4A) and one upregulated (FN1). The red and blue boxes on the right indicate the relative amounts of the corresponding protein in each group under study. Boxplots of the four differentially expressed proteins in the plasma of the R-PE and NR-PE patients: (**e**) pregnancy-specific beta-1-glycoprotein 1, (**f**) complement C4B, (**g**) complement C4A, and (**h**) fibronectin. R-PE, responsive preeclampsia.; NR-PE, nonresponsive preeclampsia.

**Figure 2 ijms-25-08738-f002:**
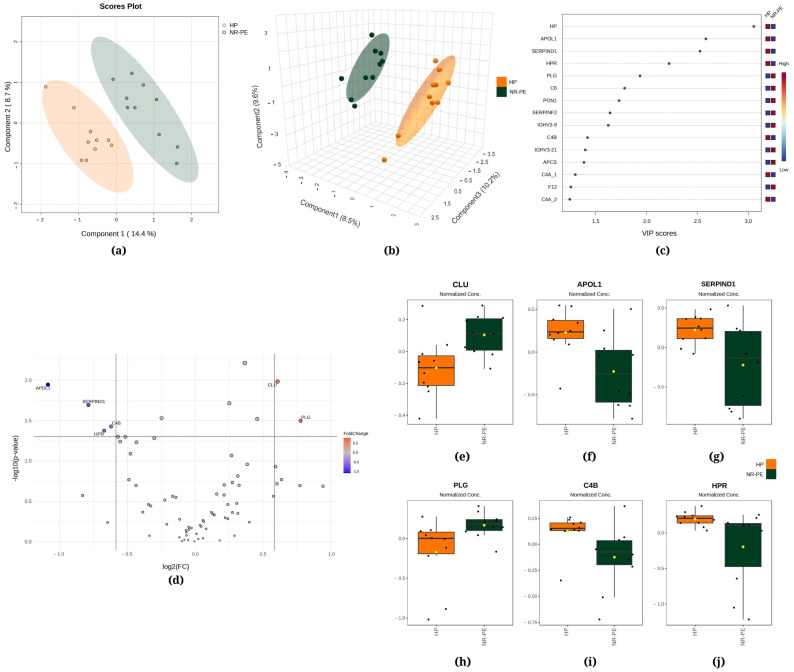
Chemometric and univariate statistical analyses of the proteins identified in the plasma of healthy pregnant (HP, in orange) women and nonresponsive preeclampsia (NR-PE, in dark green) patients. (**a**) Two-dimensional partial least squares discriminant analysis (PLS-DA) score plot. (**b**) Three-dimensional sparse partial least squares-discriminant analysis (sPLS-DA) score plot. (**c**) Variable importance in projection (VIP) represents the score of the proteins that mostly contribute to the differentiation between the HP and NR-PE groups, as identified through PLS-DA. (**d**) Volcano plot of the proteins showing that there were six differentially expressed proteins in NR-PE regarding HP: four downregulated (APOL1, SERPIND1, C4B, and HPR) and two upregulated (CLU and PLG). The red and blue boxes on the right indicate the relative amounts of the corresponding protein in each group under study. Boxplots of the six differentially expressed proteins in the plasma of HP women and NR-PE patients: (**e**) clusterin, (**f**) apolipoprotein L1, (**g**) heparin cofactor II, (**h**) plasmin heavy chain, (**i**) complement C4B, and (**j**) haptoglobin-related protein. HP, healthy pregnant; NR-PE, nonresponsive preeclampsia.

**Figure 3 ijms-25-08738-f003:**
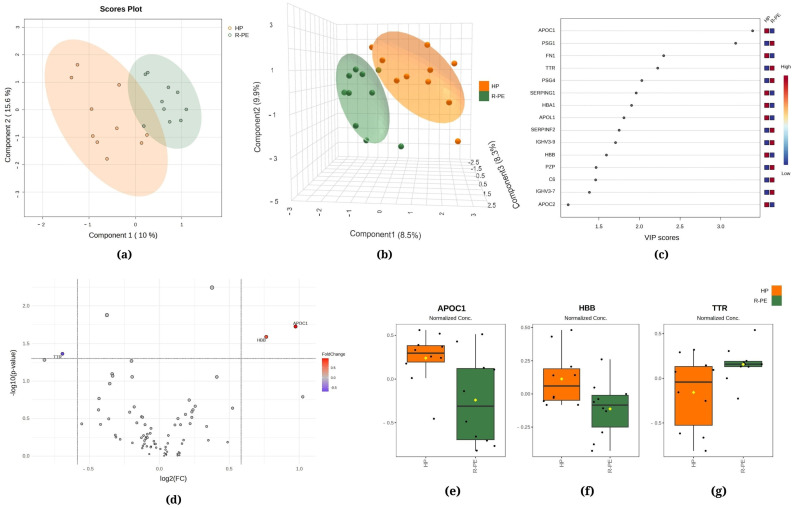
Chemometric and univariate statistical analyses of the proteins identified in the plasma of healthy pregnant (HP, in orange) women and responsive preeclampsia (R-PE, in light green) patients. (**a**) Two-dimensional partial least squares discriminant analysis (PLS-DA) score plot. (**b**) Three-dimensional sparse partial least squares-discriminant analysis (sPLS-DA) score plot. (**c**) Variable importance in projection (VIP) represents the score of the proteins that mostly contribute to the differentiation between the HP and R-PE groups, as identified through PLS-DA. (**d**) Volcano plot of the proteins showing that there were three upregulated proteins between the R-PE and HP groups: two upregulated (APOC1 and HBB) and one downregulated (TTR). The red boxes on the right indicate the relative amounts of the corresponding protein in each group under study. Boxplots of the three differentially expressed proteins in the plasma of HP women and R-PE patients: (**e**) apolipoprotein C1, (**f**) hemoglobin subunit beta, and (**g**) transthyretin. HP, healthy pregnant; R-PE, responsive preeclampsia.

**Table 1 ijms-25-08738-t001:** Clinical characteristics of the participants included in this study.

Parameters	HP	R-PE	NR-PE	*p*
Age (years)	25.1 ± 5.1	27.4 ± 4.4	31.1 ± 6.8	0.0758
BMI at blood sampling (kg/m^2^)	26.0 ± 2.6	33.7 ± 3.5 #	33.1 ± 4.2 #	0.0002
SBP (mmHg)	102.9 ± 10.3	132.0 ± 11.8 #	148.5 ± 18.5 #	<0.0001
DBP (mmHg)	67.4 ± 10.3	77.6 ± 7.4	95.8 ± 11.2 *#	<0.0001
HR (beats per minute)	81.9 ± 4.1	83.9 ± 5.8	80.3 ± 7.5	0.4895
Fasting glucose (mg/dL)	72.0 (66.8–78.0)	82.5 (78.5–128.1)	70.0 (67.0–130.5)	0.2626
Hemoglobin (g/dL)	12.1 ± 1.6	11.8 ± 1.0	11.2 ± 2.7	0.6114
Hematocrit (%)	34.7 (33.0–43.1)	35.5 (32.0–37.5)	34.8 (34.0–38.9)	0.9099
Creatinine (mg/dL)	NA	0.7 ± 0.1	0.8 ± 0.1	0.1046
Urea (mg/dL)	NA	15.4 ± 4.9	22.9 ± 9.1 *	0.0441
Proteinuria (mg/24 h)	NA	543.0 ± 171.0	2828.0 ± 2480.0	0.0721
GA at blood sampling (weeks)	37.0 (36.0–38.3)	38.0 (28.8–40.0)	33.5 (28.5–36.3)	0.0975
GA at delivery (weeks)	39.0 (39.0–40.0)	39.0 (38.0–41.0)	37.0 (29.5–37.5) *#	0.0014
Newborn weight (g)	3081.0 ± 501.7	3438.0 ± 673.6	1989.0 ± 899.3 *#	0.0005
Antihypertensive drugs taken at blood sampling				
Women taking methyldopa (%)	NA	60.0	80.0 *	0.0020
Women taking nifedipine (%)	NA	0.0	30.0 *	<0.0001
Women taking hydralazine (%)	NA	0.0	10.0 *	0.0012

BMI, body mass index; DBP, diastolic blood pressure; GA, gestational age; HR, heart rate; HP, healthy pregnant; NA, not applicable; NR-PE, nonresponsive preeclampsia; R-PE, responsive preeclampsia; SBP, systolic blood pressure. * *p* < 0.05 vs. R-PE # *p* < 0.05 vs. healthy pregnant. Data are expressed as mean ± standard deviation, median (25th–75th percentile), or percentage.

**Table 2 ijms-25-08738-t002:** Correlations between differentially expressed proteins and clinical variables in patients with preeclampsia.

Clinical and Biochemical Parameters	FN1	C4A	C4B	PSG1
SBP (mmHg)	r = −0.2752	r = −0.5199	r = −0.5193	r = −0.0529
	*p* = 0.2541	*p* = 0.0225	*p* = 0.0227	*p* = 0.8297
DBP (mmHg)	r = −0.0005	r = −0.4919	r = −0.4916	r = −0.1071
	*p* = 0.9983	*p* = 0.0324	*p* = 0.0325	*p* = 0.6626
Creatinine (μmol/L)	rs = −0.0027	rs = −0.5969	rs = −0.5833	rs = −0.2882
	*p* = 0.9912	*p* = 0.0070	*p* = 0.0088	*p* = 0.2315
Proteinuria (mg/24 h)	rs = 0.2759	rs = −0.2825	rs = −0.2909	rs = −0.1885
	*p* = 0.4082	*p* = 0.3968	*p* = 0.3863	*p* = 0.5757
SGOT (U/L)	rs = 0.4648	rs = −0.0464	rs = −0.0581	rs = −0.0716
	*p* = 0.0389	*p* = 0.8459	*p* = 0.8079	*p* = 0.7643
GA at delivery (weeks)	rs = 0.0512	rs = 0.6782	rs = 0.6775	rs = 0.6013
	*p* = 0.8400	*p* = 0.0020	*p* = 0.0020	*p* = 0.0083
Newborn Weight (g)	rs = −0.2112	rs = 0.6315	rs = 0.6316	rs = 0.6508
	*p* = 0.3713	*p* = 0.0028	*p* = 0.0028	*p* = 0.0019

C4A, complement C4A; C4B, complement C4B; DBP, diastolic blood pressure; FN1, fibronectin; GA, gestational age; PSG1, pregnancy-specific beta-1-glycoprotein; SBP, systolic blood pressure; SGOT, serum glutamic oxaloacetic transaminase. The r, rs, and *p* values are reported, where r represents Pearson’s correlation and rs indicates Spearman’s correlation.

## Data Availability

The original mass spectrometry data presented in the study are openly available in the MassIVE Repository from Computer Science and Engineering University of California, San Diego (https://massive.ucsd.edu/, accessed on 22 July 2024) with the dataset identifier: MSV000095404, and https://doi.org/10.25345/C50V89V22.

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
