# Peer review of "Different Proteomic Profiles Regarding Antihypertensive Therapy in Preeclampsia Pregnant"

_ijms, 2024, doi:10.3390/ijms25168738_

Round 1

Reviewer 1 Report

Comments and Suggestions for Authors

This manuscript describes a mass spectrometry assay for plasma protein relative quantification. With this proteomics analysis, the authors identified differentially expressed proteins between responsive (R-PE), nonresponsive (NR-PE), and healthy pregnant controls (HP), revealing potential targets for improved blood pressure control in PE. The authors use some up-to-date references to support their findings. However, there are still some aspects that are suggested to be addressed before further request:

1. this manuscript presents the MS method for the proteomic analysis, however, no MS data is provided to support the finding, the authors should consider adding some MS data to support the results, which could be considered included in SM

2. please address the conclusion section, providing a more comprehensive conclusion

3. figures need to be addressed, please use a larger font size. for the symbol size of Figure b in all figures 1/2/3 is a suitable size, the authors might want to keep the same size for Figure a/c/d. besides, figure a/c/e-j is missing the legend, what each color means.

4. the author might want to improve the writing for a better understanding

5. for references, please keep the same way, for example page 11 line 391 reference [66] the sentence is incomplete. 

Comments on the Quality of English Language

In order to have a better understanding, the scientific writing needs to be improved for this manuscript.

Author Response

We are grateful for the valuable and constructive comments provided by the reviewers on our manuscript entitled “Different proteomic profiles regarding antihypertensive therapy in preeclampsia pregnant.” We believe that their insightful feedback has significantly improved our work's quality and clarity. In the following section, we provide a point-by-point response to each comment, with the requisite revisions highlighted in yellow throughout the resubmitted manuscript.

REVIEWER #1

Comment 1: This manuscript describes a mass spectrometry assay for plasma protein relative quantification. With this proteomics analysis, the authors identified differentially expressed proteins between responsive (R-PE), nonresponsive (NR-PE), and healthy pregnant controls (HP), revealing potential targets for improved blood pressure control in PE. The authors use some up-to-date references to support their findings. However, there are still some aspects that are suggested to be addressed before further request:

  1. this manuscript presents the MS method for the proteomic analysis, however, no MS data is provided to support the finding, the authors should consider adding some MS data to support the results, which could be considered included in SM.

Response 1: We are grateful for your consideration of our study findings and your valuable suggestion regarding data availability. We acknowledge the importance of providing mass spectrometry data to support our findings and we have now uploaded the raw MS data to the MassIVE Repository from Computer Science and Engineering University of California, San Diego (https://massive.ucsd.edu/, accessed on 22 July 2024) with the dataset identifier MSV000095404, and doi: 10.25345/C50V89V22.

Comment 2: please address the conclusion section, providing a more comprehensive conclusion.

Response 2: We have revised the conclusion to provide a more comprehensive summary of our findings and the protein involvement with inflammatory, immune, and metabolic processes, as well as implications for future research and clinical applications in personalized therapy (page 12, lines 472-483).

Comment 3: figures need to be addressed, please use a larger font size. for the symbol size of Figure b in all figures 1/2/3 is a suitable size, the authors might want to keep the same size for Figure a/c/d. besides, figure a/c/e-j is missing the legend, what each color means.

Response 3: To enhance the readability and comprehensibility of the data presented, we have increased the font size and added legends in all figures. Additionally, we have reformatted Table 1 to improve its readability.

Comment 4: the author might want to improve the writing for a better understanding.

Response 4: We have undertaken a comprehensive language revision to improve the readability and overall understanding of the text. We hope the revised version meets the expected standards.

Comment 5: for references, please keep the same way, for example page 11 line 391 reference [66] the sentence is incomplete.

Response 5: We have carefully reviewed and completed the sentence associated with the reference, ensuring its clarity and coherence (page 11, lines 408-410).

Reviewer 2 Report

Comments and Suggestions for Authors

The authors have attempted to address the consequence of antihypertensive therapy applied to the case of hypertensive disorder 'preeclampsia (PE)'. This pregnancy-related disorder is associated with various organ-specific damages and cardiovascular issues. This antihypertensive therapy is claimed to be effective for only a little higher than half of the cases making it highly likely to be a case study to be thoroughly inspected. The proteomics profiling of proteins has been studied using a few sample cases. The authors have compared specifically the circulating protein profiles and attempted to identify differentially expressed proteins of 10 responsive and 10 nonresponsive patients to 10 healthy controls. Plasma protein relative quantification was performed using mass spectrometry for collecting data and then these data were used for bioinformatics analysis based on Uniprot database, PatternLab for Proteomics, and MetaboAnalyst software. They have identified a few proteins being upregulated, and a few being downregulated that are associated with PE. 

The article is well written, data are processed professionally and presented. Bioinformatics profiling is appreciable. Algorithms are well addressed.

However, a few issues have been identified as follows:

1. Introduction:  it has to be a bit rigorous, addressing the in-depth analysis of the disease, pregnancy cases, and comparative health status (specifically, whether they might experience issues related to other known or unknown complicacy, etc.) of the subsects before and during the therapy,  etc.  The Introduction should have elaborated a bit more on the proteomics software being applied to other cases that are similar to this case of study.

5. Conclusion: Vague (just one sentence containing nothing scientific).

It has to be created with a sensible amount of information.

Author Response

We are grateful for the valuable and constructive comments provided by the reviewers on our manuscript entitled “Different proteomic profiles regarding antihypertensive therapy in preeclampsia pregnant.” We believe that their insightful feedback has significantly improved our work's quality and clarity. In the following section, we provide a point-by-point response to each comment, with the requisite revisions highlighted in yellow throughout the resubmitted manuscript.

REVIEWER #2

Comment 1: The authors have attempted to address the consequence of antihypertensive therapy applied to the case of hypertensive disorder 'preeclampsia (PE)'. This pregnancy-related disorder is associated with various organ-specific damages and cardiovascular issues. This antihypertensive therapy is claimed to be effective for only a little higher than half of the cases making it highly likely to be a case study to be thoroughly inspected. The proteomics profiling of proteins has been studied using a few sample cases. The authors have compared specifically the circulating protein profiles and attempted to identify differentially expressed proteins of 10 responsive and 10 nonresponsive patients to 10 healthy controls. Plasma protein relative quantification was performed using mass spectrometry for collecting data and then these data were used for bioinformatics analysis based on Uniprot database, PatternLab for Proteomics, and MetaboAnalyst software. They have identified a few proteins being upregulated, and a few being downregulated that are associated with PE.

The article is well written, data are processed professionally and presented. Bioinformatics profiling is appreciable. Algorithms are well addressed.

However, a few issues have been identified as follows:

  1. Introduction: it has to be a bit rigorous, addressing the in-depth analysis of the disease, pregnancy cases, and comparative health status (specifically, whether they might experience issues related to other known or unknown complicacy, etc.) of the subsects before and during the therapy, etc.  The Introduction should have elaborated a bit more on the proteomics software being applied to other cases that are similar to this case of study.

Response 1: We have revised the Introduction to include a more in-depth explanation of preeclampsia and have elaborated on the use of proteomics software in similar studies.

Comment 2:

5. Conclusion: Vague (just one sentence containing nothing scientific). It has to be created with a sensible amount of information.

Response 2: We have amended the conclusion to ensure it provides a more substantial summary of our proteomic findings in the context of antihypertensive therapy in preeclampsia patients (page 12, lines 472-483).

Round 2

Reviewer 2 Report

Comments and Suggestions for Authors

My previous concerns have been addressed in this revised version. The article is now ready.